# Identification of New QTLs for Dietary Fiber Content in *Aegilops biuncialis*

**DOI:** 10.3390/ijms23073821

**Published:** 2022-03-30

**Authors:** László Ivanizs, Ilaria Marcotuli, Marianna Rakszegi, Balázs Kalapos, Kitti Szőke-Pázsi, András Farkas, Edina Türkösi, Eszter Gaál, Klaudia Kruppa, Péter Kovács, Éva Darkó, Éva Szakács, Mahmoud Said, Petr Cápal, Jaroslav Doležel, Agata Gadaleta, István Molnár

**Affiliations:** 1Department of Biological Resources, Centre for Agricultural Research, Eötvös Loránd Research Network, Brunszvik u.2, 2462 Martonvásár, Hungary; ivanizs.laszlo@atk.hu (L.I.); rakszegi.marianna@atk.hu (M.R.); kalapos.balazs@atk.hu (B.K.); pazsi.kitti@atk.hu (K.S.-P.); farkas.andras@atk.hu (A.F.); turkosi.edina@atk.hu (E.T.); gaal.eszter@atk.hu (E.G.); kruppa.klaudia@atk.hu (K.K.); kovacs.peter@atk.hu (P.K.); darko.eva@atk.hu (É.D.); molnar.istvan@atk.hu (I.M.); 2Department of Agricultural and Environmental Science, University of Bari ‘Aldo Moro’, Via G. Amendola 165/A, 70126 Bari, Italy; ilaria.marcotuli@uniba.it (I.M.); agata.gadaleta@uniba.it (A.G.); 3Institute of Experimental Botany of the Czech Academy of Sciences, Centre of the Region Haná for Biotechnological and Agricultural Research, Šlechtitelů 31, 779 00 Olomouc, Czech Republic; said@ueb.cas.cz (M.S.); capal@ueb.cas.cz (P.C.); dolezel@ueb.cas.cz (J.D.); 4Field Crops Research Institute, Agricultural Research Centre, 9 Gamma street-Giza, Cairo 12619, Egypt

**Keywords:** *Aegilops biuncialis*, dietary fiber, β-glucan, DArTseq analysis, genome-wide association study (GWAS)

## Abstract

Grain dietary fiber content is an important health-promoting trait of bread wheat. A dominant dietary fiber component of wheat is the cell wall polysaccharide arabinoxylan and the goatgrass *Aegilops biuncialis* has high β-glucan content, which makes it an attractive gene source to develop wheat lines with modified fiber composition. In order to support introgression breeding, this work examined genetic variability in grain β-glucan, pentosan, and protein content in a collection of *Ae. biuncialis*. A large variation in grain protein and edible fiber content was revealed, reflecting the origin of *Ae. biuncialis* accessions from different eco-geographical habitats. Association analysis using DArTseq-derived SNPs identified 34 QTLs associated with β-glucan, pentosan, water-extractable pentosan, and protein content. Mapping the markers to draft chromosome assemblies of diploid progenitors of *Ae. biuncialis* underlined the role of genes on chromosomes 1M^b^, 4M^b^, and 5M^b^ in the formation of grain β-glucan content, while other QTLs on chromosome groups 3, 6, and 1 identified genes responsible for total- and water-extractable pentosan content. Functional annotation of the associated marker sequences identified fourteen genes, nine of which were identified in other monocots. The QTLs and genes identified in the present work are attractive targets for chromosome-mediated gene transfer to improve the health-promoting properties of wheat-derived foods.

## 1. Introduction

Bread wheat (*Triticum aestivum* L., 2*n* = 6*x* = 42; AABBDD) is grown on more than 220 million hectares worldwide and is one of the most important components of the human diet. Due to its major role in human nutrition, wheat is a major source of dietary fiber (DF). DF has numerous health-promoting effects, including lowering the risk of coronary heart disease, colorectal cancer, inflammatory bowel disease, breast cancer, tumor formation, and mineral-related abnormalities [1]. The major DF components of the wheat grain are the cell wall polysaccharides, arabinoxylan (AX) and (1-3)(1-4)-β-D-glucan (β-glucan, BG), which account for about 70 and 20%, respectively, of the total cell wall polysaccharides in the starchy endosperm (and hence in white flour) [2]. DF components of wheat also affect bread-making quality, gluten-starch separation, livestock feed quality, and fermentation during the production of alcohol for beverages and biofuel [3,4,5].

Numerous studies on the amount and composition of DF in wheat and other cereals concluded that the typical component of DF in wheat and rye is AX, while the grain endosperm of barley and oats is rich in β-glucan [6,7,8,9,10,11]. Because of the significant effects on human health and processing characteristics, it is desirable to modify the content and composition of DF in wheat. The development of wheat varieties with altered AX vs. β-glucan ratio is an attractive strategy to produce functional foods. However, during thousands of years of domestication and cultivation, the genetic diversity of bread wheat narrowed down [12] and it is challenging to discover suitable allelic variations for the genes that change the composition of DF.

The genus *Aegilops* (goatgrass) is closely related to *Triticum* and includes eleven diploid and twelve polyploid species with divergent U, M, N, C, S, D, and T genomes. Numerous reports pointed to wild goatgrasses as useful and untapped sources of new gene variants for wheat improvement [13,14,15,16]. Allotetraploid *Ae. biuncialis* Vis. (2*n* = 4*x* = 28; U^b^U^b^M^b^M^b^) evolved after natural hybridization between *Ae. umbellulata* Zhuk. (2*n* = 2*x* = 14; UU) and *Ae. comosa* Sm. in Sibth. et Sm. (2*n* = 2*x* = 14; MM), which served as U- and M genome donors. *Ae. biuncialis* is native from the Iberian Peninsula to Central Asia and includes the entire Mediterranean region, Near East, Cis- and Transcaucasia, and the southern part of Ukraine [13,17]. In parallel with its wide distribution and adaptation to different agro-climatic conditions, *Ae. biuncialis* exhibits high genetic diversity [18]. Some genotypes were found to be highly resistant to important cereal diseases such as barley yellow dwarf luteovirus [19], powdery mildew [20], yellow rust [21], brown rust [22], leaf rust [23], and stem rust [24], while others have a high tolerance to abiotic stresses such as frost [25], salt [26,27], and drought [28,29]. *Ae. biuncialis* grains have also been found to contain a significantly larger amount of micronutrients [30], edible fiber, and protein [31,32,33] compared to bread wheat. To introduce favorable agronomic traits of *Aegilops*, some U^b^- and M^b^-genome chromosomes were transferred into wheat by the development of wheat-*Ae. biuncialis* amphiploids, disomic addition lines carrying chromosomes 1U^b^, 2U^b^, 3U^b^, 2M^b^, 3M^b^ and 7M^b^, a double disomic addition line 1U^b^ and 6U^b^, 3M^b^(4B) substitution, and 3M^b^.4BS translocation lines [30,34,35,36,37,38].

Although numerous wheat-alien addition and translocation lines have been developed, the genetic potential of *Ae. biuncialis* remains unexploited in wheat breeding [14]. Chromosome-mediated transfer of grain dietary fiber content and composition into wheat has been hampered by poor knowledge on the intraspecific variation and genomic regions determining these traits in *Ae. biuncialis*. For example, biochemical analysis of wheat-*Aegilops* addition lines showed that chromosomes 5U^g^, 7U^g^, 7M^g^, and 7M^b^ of *Ae. geniculata* or *Ae. biuncialis* have a positive effect on DF content of bread wheat [31]. However, the β-glucan content of these addition lines was only 20–30% higher than that of the wheat parent and remained low compared to the values of the *Aegilops* parents (3-500% β-glucan content compared to wheat). These results suggested the presence of additional QTLs in the U and M genome chromosomes, which are needed to further increase the β-glucan content of wheat. Examination of β-glucan content in a diverse collection of *Ae. biuncialis* accessions would facilitate the selection of the most suitable crossing partners for introgression breeding programs and allow mapping of the additional QTL regions and candidate chromosomes for targeted gene transfer into wheat.

Diversity Array Technology (DArT), originally developed as a hybridization-based microarray platform, is a sequence-independent, cost-effective, high-throughput genotyping-by-sequencing approach combining the genome complexity reduction method and next-generation sequencing [39,40,41]. DArTseq platform can be used to genotype wild and cultivated species, generating thousands of two types of molecular markers simultaneously. Dominant Silico-DArT markers are genomic sequences representing the presence–absence variation of the respective fragment, whereas the co-dominant SNP-DArT markers indicate nucleotide polymorphism within the fragment. Both types of markers provide dense coverage of the entire genome and enable efficient detection of QTLs. DArTSeq platform has already been used for whole-genome profiling in *Aegilops* species, including *Ae. taushii* [42] and *Ae. sharonensis* [43].

Ivanizs et al. [18] characterized phenotypic variation in heading time and genetic diversity in a collection of *Ae. biuncialis* accessions with geographically diverse origins using Silico-DArT markers. More recently, the complete chromosome set of *Ae. umbellulata* and *Ae. comosa* [44] was flow-sorted and sequenced by Illumina technology to generate chromosome-specific de novo draft sequence assemblies for the U- and M-genomes [45]. The chromosome-specific genomic resources thus obtained allowed us to determine the chromosomal location of orthologous genes and to develop markers for *Aegilops* chromosomes [45].

Motivated by the need to discover accessions suitable for interspecific hybridization programs and to identify QTLs for targeted transfer of high fiber and/or protein content, this work focused on the evaluation of phenotypic variability of β-glucan, total- and water-extractable pentosan, and protein content in a collection of 86 genotypes of *Ae. biuncialis*. Using the allelic composition of DArTSeq derived SNP markers in combination with a sequence similarity search on the U- and M-chromosome assemblies, we characterized the *Aegilops* genotypes to find markers and genomic regions closely associated with the desired quality traits.

## 2. Results

### 2.1. Characterization of Quality Traits

Grain samples from single plots were analyzed to determine the variation in dietary fiber components, β-glucan (BG) and arabinoxylan, measured as total- (TP) and water-extractable pentosans (WEP) as well as the protein content of *Ae. biuncialis* accessions and the control wheat line Mv9kr1. As three *Aegilops* accessions produced a low quantity of grains, the quality traits were only determined in 83 genotypes. The accessions were chosen to represent a wide geographical distribution of *Ae. biuncialis*, extending from the Balkans to the Caspian Sea. As a result, a large variation was found for all traits that were analyzed.

The BG content of the *Aegilops* collection was significantly higher than that of the Mv9kr1 wheat (Table 1, Appendix A), with the *Aegilops* accessions exhibiting an average 38.1 mg/g BG content, ranging between 22.7 mg/g and 54.9 mg/g, and Mv9kr1 wheat line exhibiting much lower value (9.44 mg/g).

Although there was no difference in total pentosan content between the *Aegilops* accessions (40.11 mg/g) and wheat (40.17 mg/g), the *Aegilops* collection showed a wide phenotypic variation (29.60 mg/g–50.77 mg/g). While the average content of water-extractable pentosan in *Aegilops* genotypes (10.82 mg/g) did not differ considerably from that of Mv9kr1 wheat (10.79 mg/g), a wide range (6.83 mg/g–15.42 mg/g) was observed in *Ae. biuncialis*. In terms of protein content, Mv9kr1 wheat line (12.91%) was significantly outperformed by all *Ae. biuncialis* accessions (by a mean value of 26.61% and a range from 19.61% to 33.49%). In summary, all *Ae. biuncialis* genotypes had significantly higher grain BG and protein content than Mv9kr1 wheat, as well as a high variation in the pentosan fractions.

To better understand the effect of genotype and environmental factors on grain composition, the two-year dataset of the 83 *Ae. biuncialis* accessions were investigated. Each year, the frequency distribution of the traits was studied separately. Because all of the analyzed traits had a normal distribution, their polygenic nature was confirmed (Appendix A). Although genotype (G) and environment (E) had a significant effect on β-glucan content (Table 2), the G factor explained a greater proportion (77%) of the variance (Figure 1). This was supported by its high heritability (0.93).

The TP content did not differ significantly between seasons, indicating that the environment had no significant effect on this trait (Table 2). On the other hand, genotype had a strong influence on the TP and WEP content, accounting for approximately one-third of the total variance, 37% and 43%, respectively (Figure 1). Both pentosan fractions were found to have a high degree of heritability (0.54 and 0.61, respectively).

The analysis of protein content revealed that the G × E interaction determined 83% of the total variance (Figure 1), indicating that *Aegilops* genotypes responded differently to different environmental conditions. That is most likely the reason for the low heritability (0.27) values and the fact that G accounted for only 16% of the total variance. Nonetheless, for the genome-wide association analysis, all of the quality traits were used.

### 2.2. Population Structure

After excluding accessions with failed DNA amplification, monomorphism, missing data, and allele frequencies that differed from what was described in the methods section, a total of 2602 DArTSeq derived SNP tags were used for whole-genome profiling (the population analysis and the GWAS) in the set of 86 *Ae. biuncialis* accessions.

To classify genotypes into subpopulations based on genetic similarity, the relatedness of the *Aegilops* accessions was studied using the Bayesian approach implemented in STRUCTURE, which provided the ΔK plotted against the K numbers of the subgroups. The optimal number of subpopulations was determined using the STRUCTURE Harvester software. The maximum ΔK (686.70) occurred at K = 2 (Figure 2), resulting in the identification of two groups in the *Aegilops* collection. Based on DArTSeq-derived SNP data, *Ae. biuncialis* collection was divided into two subpopulations: subpopulations 1 and 2 (Figure 3).

The majority of *Aegilops* genotypes (63 accessions) were assigned to subpopulation 1, with a Q1 mean membership of 0.85. The remaining 23 accessions were assigned to subpopulation 2, with a Q2 mean of 0.96. Based on the genetic data of the SNP-DArT markers, an unrooted phylogenetic tree was constructed using the maximum likelihood method to group accessions into clades (Figure 4). The accessions of *Ae. biuncialis* were classified into two main subpopulations, which corresponded to the grouping pattern obtained from the STRUCTURE analysis (Figure 3 and Figure 4).

To identify the genetic structure within the *Ae. biuncialis* population, principal coordinate analysis (PCoA) (Figure 5) was performed on the dataset of SNP-DArT markers obtained from 86 *Aegilops* genotypes. The first three coordinates explained 14.5%, 7.1%, and 4.8% of the variation, respectively, accounting for a total of 26.4%. The collection of *Ae. biuncialis* could be divided into two major groups, which fitted well with the subpopulations revealed by the Bayesian statistical method.

The three statistical approaches yielded consistent results for the genetic diversity of *Ae. biuncialis* population, in which the *Aegilops* genotypes were grouped into two subpopulations based on their geographic distribution. The accessions of *Ae. biuncialis* in subpopulation 1 come from the Peloponnese, Asia Minor, Azerbaijan, and the Near East, while the 23 members of the other subpopulation originate from the Balkans and North Africa (Figure 6).

### 2.3. Determination of Marker-Trait Associations

Genome-wide association study (GWAS) was conducted during two seasons using the MLM (Q+K) model, allowing the identification of 34 QTLs representing regions associated with β-glucan, total-pentosan, water-extractable pentosan, and protein content (Table 3). As three accessions could not be characterized for quality traits, 83 accessions were analyzed by GWAS. Primary to the GWAS analysis, the estimation of heredity divergence within the intended germplasm was determined through linkage disequilibrium (LD) calculation. LD estimated using the squared-allele frequency correlations (r^2^), and based on genome-wide DArTSeq derived SNP markers was low with a mean r^2^ = 0.049. The information about the LD of DArTSeq derived SNP markers has been presented in Appendix A. A marker-trait association (MTA) was considered significant when one or more markers had a LOD value greater than three. QQ plot and Manhattan plot were generated for individual traits based on the MLM and reported in Appendix A, respectively. WEP content was found to be involved in most MTAs, followed by grain protein content, whereas BG and TP contents were involved in three MTAs. Some QTLs, such as QTL-1, 2, 4, 5, 7, 8, and 12, were linked to multiple traits, and the multiple associations involved protein and fiber traits.

### 2.4. Chromosomal Location and Functional Annotation of the QTLs

To determine the chromosomal location of the marker-trait associations, sequences of thirty-four SNP-DArT markers were aligned to the wheat pseudomolecules (Appels et al., 2018) and to the assemblies of chromosomes 1U-7U and 1M-7M, which were obtained by sequencing chromosomes flow-sorted from diploid *Ae. umbellulata* and *Ae. comosa*, respectively [45]. Using two best hits, we determined chromosomal positions of 32 markers in *Aegilops* U or M genomes and hexaploid wheat A, B, and D genomes. Twenty-five (78.1%) of the 32 markers were found on the same homoeologous group chromosomes in *Aegilops* as in wheat (Table 3).

A sequence similarity search was also performed using the sequences of SNP-DArT markers to predict genes associated with MTAs (Appendix A). Wheat gene sequences were obtained from the GrainGenes database (https://wheat.pw.usda.gov, accessed on 30 October 2021) and used for the NCBI BLAST search. The analysis identified fourteen genes, nine of which encode putative ripening-related protein 6 associated with the QTL for WEP and protein; 1-deoxy-D-xylulose-5-phosphate synthase 1, glutamate receptor 2.8-like, endoglucanase 5-like, and flavonol synthase/flavanone 3-hydroxylase-like, all co-located with WEP QTL; glutathione S-transferase 3-like linked to β-glucan, WEP and protein loci (Table 3).

## 3. Discussion

Alien introgression breeding is a promising approach to modifying the dietary fiber composition of bread wheat. In this work, we used GWAS for the first time to identify QTL regions determining dietary fiber composition in the wild wheat relative *Ae. biuncialis*. Suitable crossing partners with superior compositional traits, trait-related genomic data, and molecular tools obtained from the diversity scan of dietary fibers in a DArTSeq-characterized collection of *Ae. biuncialis* accessions will facilitate the chromosome-mediated improvement of bioactive components in wheat.

### 3.1. Grain Composition of Ae. biuncialis

Grain β-glucan, TP, WEP, and protein content in *Ae. biuncialis* accessions and Mv9kr1 wheat line as determined in the present study were similar to those estimated earlier when the effect of added *Aegilops* chromosomes on the grain composition of wheat was analyzed [31,33]. Diversity analysis of 43 wild and cultivated diploid, tetraploid and hexaploid wheat accessions demonstrated that β-glucan is the dominant form of cell wall polysaccharides in *Aegilops* species with U and M genomes.

We found a considerable level of variation in the compositional traits of the *Aegilops* collection, which enabled association analysis and mapping. The analysis of variance revealed that the G was the main factor controlling most of the studied traits. However, the G × E studies found that traits related to cell wall polysaccharides, particularly the β-glucan content, had a high level of heritability, making these traits suitable for identifying marker-trait associations via GWAS analysis. On the other hand, protein content had low heritability, implying that the marker-trait associations found for this parameter could be misleading and need to be supported by future experiments.

There have been numerous studies on the effects of G and E on the edible fiber content of common wheat [46,47,48,49], but only a limited amount of information on the variation of grain compositional traits in *Aegilops* species has been published. Marcotuli et al. [32] investigated grain BG content and kernel weight in a panel of *Triticum* and *Aegilops* species. Similar to our study, a two-year analysis revealed a dominant role of the genotype in the variation of BG content.

Our systematic analysis for grain compositional traits identified *Ae. biuncialis* accessions with high BG content. The accessions TA2074 and MvGB381 had the highest level of BG, about 500% higher than the BG level of the control wheat line, pointing to these *Ae. biuncialis* genotypes as crossing partners of choice for chromosome-mediated modification of dietary fiber composition in bread wheat.

### 3.2. Genetic Diversity of the Ae. biuncialis Collection

There are only a few reports on the genetic diversity of *Ae. biuncialis*. Based on AFLP markers, *Ae. biuncialis* accessions from the two regions of the Iberian Peninsula were clustered in two groups, which correlated with their geographic origin [50]. *Ae. biuncialis* genotypes from the eastern side of the Black Sea and the western coast of the Caspian Sea were classified into two subgroups using RAPD markers, which corresponded with the clusters from the two areas of Transcaucasia [51]. Genetic diversity of *Ae. geniculata*, another tetraploid U- and M-genome species closely related to *Ae. biuncialis*, was characterized by AFLP markers [52], where a complete set of 411 *Ae. geniculata* individuals were grouped into seven subpopulations with a high level of correlation with geographic distribution in the Mediterranean Basin.

Using the Bayesian approach and PCoA for the SNP-DArT marker data, the present study identified two subpopulations with different geographic distributions in the *Ae. biuncialis* collection: subpopulation 1 comprised 63 genotypes from the Peloponnese, Asia minor, Azerbaijan, and the Near East, while subpopulation 2 originated primarily from the Balkans and North Africa. An earlier diversity analysis of the same *Ae. biuncialis* collection made by 32,700 dominant Silico-DArT markers identified five groups (A-E), which correlated with their geographic origin: A (North Africa), B (Balkans and West Turkey), C (Balkans and Near East), D (Peloponnese, Asia Minor, and Crimea) and E (Near East and Azerbaijan) [18]. Although a different number of subpopulations was identified depending on the type of markers used (Silico- vs. SNP-DArTseq), the spatial pattern of genetic diversity of *Ae. biuncialis* population showed high similarity. Based on co-dominant SNP markers, the accessions assigned previously to clusters D and E were grouped into subpopulation 1, while the genotypes belonging to clusters A and B were classified as subpopulation 2 in the present study (Figure 5). Genotypes of *Ae. biuncialis* originating from North Africa and the Balkans (former subpopulations A and B) were clustered into one group (subpopulation 2) in the present study, suggesting that these genotypes share a common phylogenetic origin. Subpopulation 1 included genotypes previously classified into clusters D (Peloponnese, Asia Minor, and Crimea) and E (Near East and Azerbaijan) also indicating a closer phylogenetic relationship. A majority of *Aegilops* accessions from former cluster D have a mixed genome shared in different proportions by subpopulations 1 and 2, indicating a significant admixture of their ancestral genotypes. Because genotypes in subpopulation 1 are mostly found in southern Greece and Turkey, while most of the accessions in subpopulation 2 are originating from the Balkans and the western part of Turkey, their natural distribution areas may overlap in the Aegean region and Asia Minor, allowing intraspecific hybridizations to occur. The analysis of chloroplast DNA could provide more information about the genealogical relationship of subpopulations and the intraspecific evolution of *Ae. biuncialis* [53].

Pairwise LD measured by r^2^ based on DArTSeq derived SNP markers was low (mean r^2^ = 0.049). This is expected as in self-pollinated species, such Arabidospis or barley, LD can extend up to 10–30 kb and 212 kb, respectively [54,55]. LD, in fact, results from the interplay of many factors, such as selection, which causes locus-specific bottlenecks and it is one of the factors that increase LD between selected alleles at linked loci [56]. LD estimates provided here suggest the suitability of the markers used to perform GWAS and the success of the analysis.

### 3.3. GWAS and Candidate Gene Identification

Using the allelic distribution of SNP markers in the collection of *Ae. biuncialis*, 34 marker-trait associations were found, identifying QTL regions for grain BG, TP, WEP, and protein content. The present study also demonstrated the utility of chromosome genomics in facilitating genome analysis of wild gene source species with poorly studied genomes [57]. Because the segregating genetic map for *Ae. biuncialis* was not available, we used a sequence similarity search against the assemblies of individual chromosomes from the U and M genomes of *Ae. umbellulata* and *Ae. comosa*, respectively [45], to locate the associated markers on chromosomes of *Aegilops*. However, the chromosomal position of the QTLs established could be affected by the purity of the flow-sorted M-genome chromosome fractions that were sequenced [44,45].

The present study identified five QTLs (QTL3, 6, 16, 23, and 34) on *Aegilops* chromosomes 1M and 4M, while the same markers were found on wheat group 1 chromosomes. Previous comparative studies of wheat and diploid *Aegilops* species based on the mapping of conserved orthologous genes using single-gene FISH [45] and COS markers [44,58] revealed high macrosynteny of M-genome chromosomes with wheat homoeologs. Identification of the associated markers on different homoeologous groups in *Ae. comosa* (4M) and wheat (group 1) could be due to cross-contamination of chromosome 4M fraction by 1M chromosomes during flow-sorting. This explanation is supported by the fact that the 4M and 1M chromosome populations were detected close to each other on the bivariate flow karyotype of *Ae. comosa* [44,45]. Thus, we concluded that the most likely chromosome location of these markers was on the M-genome chromosome, which corresponded to the wheat homoeologous group identified by the BLAST analysis.

Wheat-*Ae. umbellulata* synteny should also be considered when determining the chromosomal position of marker-trait associations. The U genome of *Ae. umbellulata* has been reported to have undergone multiple genome rearrangements during the evolution, resulting in synteny breaks in some chromosomes relative to wheat [44,58,59,60,61]. Comparative genome analysis using segregating mapping populations [59,60,61], COS markers on wheat-*Aegilops* chromosome addition lines [58], and physical mapping by single-gene FISH [45] revealed significant synteny of chromosomes 1U, 2U, 3U, and 5U with the corresponding wheat chromosomes. On the other hand, chromosome 4U contains genomic regions homoeologous with wheat (w) chromosome groups 4, 5, and 6, 6U with w1, w2, w4, w6, and w7, while 7U with w7 and w3 chromosomes [45,58,59,60,61].

We identified 30 QTLs for traits related to the cell wall polysaccharide (BG, TP, and WEP) synthesis that were evenly distributed across all M genome chromosomes (1M:5, 2M:5, 3M:7, 4M:4, 5M:2, 6M:6, and 7M:1). These findings are partially consistent with the previous biochemical analysis of wheat-*Ae. geniculata* and wheat-*Ae. biuncialis* addition lines [31].

We mapped three QTLs related to grain BG content on chromosomes 1M, 4M, and 5M. Although chromosome 4M was not represented in the set of addition lines previously investigated by Rakszegi et al. [31], their results indicated that *Aegilops* group 5 chromosomes significantly affects cell wall polysaccharide synthesis and can modify the grain BG, TP, and WEP content in wheat. Our results suggest that genes on chromosomes 1M and 4M may also be required for BG synthesis in *Ae. biuncialis*, and that a transfer of chromosome 5M alone is insufficient to increase the wheat BG content to that of *Aegilops*.

Two (QTLs 2 and 3) of the three QTLs associated with BG content were identified on the *Aegilops* chromosomes of 1M and 5M, which is consistent with previous studies made on durum wheat [62] indicating the group 1 and 5 chromosomes have also QTLs affecting BG synthesis. Moreover, two members of the cellulose synthase-like gene family (*CslF9* and *CslF7*) involved in β-glucan biosynthesis were assigned to the same homoeologous group chromosomes (groups 1 and 5) in barley and hexaploid wheat [31,63,64]. The third marker-trait association (QTL 1) was localized on chromosome 4M. Because *Csl* gene family members have not been identified on group 4 chromosomes in barley or wheat, it is less likely that QTL 1 on the structurally similar chromosome 4M contains a *Csl* gene variant in *Ae. biuncialis*. On the other hand, functional annotation revealed that the marker associated with QTL 1 co-located with a glutathione S-transferase 3-like gene (*GTS3L*), which is involved in various metabolic pathways with a wide range of substrates, including α,β-unsaturated carbonyls [65]. As the same QTL was not only associated with the BG level but also protein and WEP content, we may conclude that QTL1 affects the metabolism of cell wall polysaccharides in general.

The water-extractable pentosan content was associated with the highest number of QTLs (24 of the 30) related to the cell wall polysaccharide synthesis. Among them, 11 QTLs affected positively the edible fiber content, with group 1 (2 QTLs), group 3 (4 QTLs), group 6 (4 QTLs), and group 4 (1 QTLs) chromosomes being over-represented. This is consistent with earlier findings in durum and bread wheat. Thus, a QTL strongly affecting the WEP content was found on chromosome 1B of hexaploid wheat [48], which represented 59% of its phenotypic variance [66]. Using genome-wide association and meta-QTL analyses on multiple mapping populations seven QTLs for grain dietary fiber content were identified on chromosomes 1B, 3A, 3D, 5B, 6B, 7A, and 7B in durum- and bread wheat [62,67]. The transcriptome analysis identified 73 candidate genes that control the edible fiber content of bread wheat. Three major genes responsible for grain fiber content were detected on chromosomes 1B, 3A, 3D, and 6B showing co-localization with seven previously identified QTLs [67].

Glycosyltransferase gene families (*GT*) play an important role in the regulation of AX biosynthesis [68], with homoeoalleles of the *GT47* gene encoding a subunit of β-1,4 xylan synthase located on the group three chromosomes in hexaploid wheat [69,70], and two alleles of *GT61* encoding α-1,3-arabinosyltransferase located on chromosomes 1B and 6A [71]. The presence of putative orthologs of wheat genes (*GT47* and *GT61*) on the 1B, 3ABD, and 6A chromosomes correlated well with the current result that the majority of QTLs associated with WEP content were identified on the same homoeologous groups of *Aegilops* chromosomes.

Our BLAST analysis identified several genes in QTLs associated with pentosan content. Five QTLs on chromosomes 1M, 3M, and 6M co-located with candidate genes encoding enzymes directly involved in carbohydrate or pentosan metabolism: endoglucanase 5-like, which hydrolyzes substrates with (1-->4)-beta-glucosyl linkages, such as xyloglucan [72]; soluble starch synthase, one of the enzymes in the starch biosynthetic network [73], glycoside hydrolase/deacetylase, catalyzing glycolytic cleavage of O-glycosidic bonds, particularly those between two glucose residues in some polysaccharides [74]; glycosyltransferase, a key enzyme involved in the production of diverse carbohydrate-containing structures, and cell surface glycans [75].

The association of QTLs related to WEP content and candidate genes influencing polysaccharide metabolism emphasizes the important role of genes on chromosomes 1M, 3M, and 6M in the biosynthesis of cell wall components in *Ae. biuncialis.*

## 4. Materials and Methods

### 4.1. Plant Material

A set of 86 *Aegilops biuncialis* Vis. (2*n* = 4*x* = 28, U^b^U^b^M^b^M^b^) accessions, originating from a wide range of ecological habitats and representing the geographic distribution of the species, was previously genotyped using the DArTSeq platform [18]. The accessions were provided by various germplasm collections across the world, as previously detailed by Ivanizs et al. [18]. The *Aegilops* accessions were maintained and propagated by the Cereal Genebank in Martonvásár. Hexaploid winter wheat (*Triticum aestivum* L.) line Mv9kr1, which contains a recessive crossability gene *kr1* [76], was used as a crossing partner in the introgression breeding programs at Martonvásár [77] and was also used in this work.

### 4.2. Field Trial

The diverse collection of *Ae. biuncialis* was grown during two growing seasons (2015–16 and 2016–17) in an experimental area (Breeders nursery, Martonvásár, Hungary, geographic coordinates: 47°19′39″ N, 18°47′01″ E) owned by the Agricultural Institute, Centre for Agricultural Research. Each genotype was sown in 6 × 3 m rows (50 grains per row, 0.15 m row spacing) in three replicates (i.e., 2 rows per replicate) using randomized complete blocks design in chernozem soil as previously described by Ivanizs et al. [18]. *Ae. biuncialis* accessions were harvested manually in each replicate, where the grain mixture of replicates 1 and 2 or replicates 2 and 3 were considered as two biological samples for each genotype.

### 4.3. Quantitative Determination of Grain Composition

Grain composition analysis was performed on grain samples from two growing seasons (2015–16 and 2016–17). Since three genotypes could not be maintained in field trials, 83 *Ae. biuncialis* accessions were characterized for quality traits (protein, β-glucan, and pentosans) together with the wheat genotype Mv9kr1. For each genotype, measurements were performed on two biological and two technical replications. Whole grain flour was prepared from 4 grams of grain per sample using a Retsch Mixer Mill MM 400 ball mill (Retsch, Haan, Germany). Wholemeal samples were immediately refrigerated and stored at −20 °C before use. Crude protein content was determined by the Dumas method in accordance with ICC Standard Method No. 167 [78] using the Elementar Rapid N III instrument (Elementar Analysensysteme GmbH, Langenselbold, Germany). Arabinoxylan content, measured as total pentosans (TP) and water-extractable pentosans (WEP), was determined using the colorimetric method following Douglas [79] with minor modifications [80], as reported by Rakszegi et al. [33]. Total mixed-linkage β-glucan content was determined in wholemeal samples using a Megazyme kit (Megazyme, Bray, Ireland) (ICC Standard Method No. 166 [81]; AACC Method No. 32-23.01 [82] as described by Rakszegi et al. [33]. Raw data for edible fiber and protein content measurements are included in Appendix A.

### 4.4. QTL and Candidate Gene Detection

Genomic DNA was extracted from young leaves of 86 accessions of *Ae. biuncialis* using a QuickGene-Mini80 device (FujiFilm, Osaka, Japan) and a QuickGene DNA tissue kit (FujiFilm) according to the manufacturer’s instructions. DNA samples of the *Ae. biuncialis* accessions were genotyped utilizing the DArTseq™ 1.0 array platform by Diversity Arrays Technologies Pty. Ltd. (University of Canberra, Australia). A total of 77,376 SNP-DArT markers were obtained after genome complexity reduction and subsequent SNP calling. Before performing GWAS, markers with a minimum allele frequency of less than 10% and those with more than 5% missing data points were removed from the data matrix using GenAlEx [83,84]. After filtering marker data, a total of 2602 SNPs were obtained and used for population structure analysis.

To determine the population structure, three different approaches were used: a principal coordinate analysis (PCoA) was performed using GenAlEx [83,84]; Bayesian clustering program STRUCTURE version 2.3.4 [85,86,87,88] was used to determine the population structure per se, using an admixture model with correlated allele frequencies, with the number of subpopulations (K) from 1 to 10, burn-in period of 10 × 1003 iterations followed by 10 × 1003 Markov chain Monte Carlo (MCMC) iterations; unrooted Bayesian tree was created with TASSEL 5.2.70 software [89] using the bootstrap method. TASSEL 5.2.70 [89] was also used to identify significant associations with the grain quality traits. GWAS was carried out using Mixed Linear Model (MLM), which accounts for the Q matrix and kinship matrix (K). MLM—(Q + K), correlation coefficient (R2), and marker effect of each SNP associated with β-glucan, pentosan, water-extractable pentosan, and protein content were estimated. To provide the adjusted p values, the false discovery rate (FDR) was calculated, using a threshold of <5%, with the q-value package [90] in the R version 3.1.1 [91].

### 4.5. Sequence Similarity Analysis and Functional Annotation

In order to obtain the chromosomal location of the thirty-four SNP markers closely associated with the quality traits, the SNP marker (35–69 bp) sequences were used as queries for the BLASTn search on 1M-7M and 1U-7U chromosomal scaffolds of *Ae. comosa* MvGB1039 (https://doi.org/10.5061/dryad.wpzgmsbk9, accessed on 21 May 2021) and *Ae. umbellulata* AE740/03 (https://doi.org/10.5061/dryad.70rxwdbwc, accessed on 21 May 2021), respectively [45], and against the reference pseudomolecules of hexaploid wheat (Ensembl Plants, release-46, [92]) using BLASTn package of the Blast Command Line Application 2.9.0 (ftp://ftp.ncbi.nlm.nih.gov/, accessed on 21 May 2021). Alignments that met certain criteria (e-value: 1e–5, max target seqs2: -max hsps1), were considered significant. The two best alignment hits were determined for each SNP marker and summarized in Appendix A.

Functional annotations were performed using translated sequences from known wheat genes. Gene Ontology information was obtained from the Universal Protein Resource (ftp://ftp.uniprot.org, accessed on 21 May 2021; UniProt release 2019_11) database, and protein domains were identified using the Hidden Markov Model (HMM)-based HMMER 3.0 software package (http://eddylab.org/software/hmmer/, accessed on 21 May 2021) [93] in Pfam 32.0 entries (ftp://ftp.ebi.ac.uk, accessed on 21 May 2021) [94]. The results of BLASTn search against the chromosome sequences of wheat, *Ae. comosa*, and *Ae. umbellulata*, the protein collection extracted from the database (PFAM), and gene ontology information (GO) are summarized in Appendix A. Finally, using the SNP sequences BLASTed against the monocot sequences annotated in NCBI, putative candidate genes associated with BG, TP, WEP, and protein content were identified. To identify *Aegilops* sequences, each “gene” or “gene family” name was searched for in the CAZy database [95]. To characterize variation in expression data in different experiments, all retrieved protein sequences were BLASTed against the PlantGDB database (http://www.plantgdb.org/cgi-bin/prj/PLEXdb/ProbeMatch.pl, accessed on 21 May 2021).

### 4.6. Statistical Analyses

Grain quality trait parameters were measured with two replications in two parallel samples per accession. The measurement was repeated two more times when the difference between the results of duplicated analyses exceeded 10%. For statistical evaluation, Linear Mixed Model analysis (using the restricted likelihood algorithm, REML) was applied for each compositional quality trait using SPSS 16.0 software (SPSS Inc., Chicago, IL, USA) as described by Virk et al. [96]. For all genotypes (G), two growing periods were considered as environments (E), with the E and the G being fixed factors. Multiple measurements of each genotype (two parallel samples with two replications) were regarded as the random factor (R). For each quality parameter, genotypic variance, repeatability, and variance of G × E interaction were calculated. The broad-sense heritability was evaluated based on the proportion of genotypic to phenotypic variance (h2 = VG/VP) using the following formula: VP = VG + (VG × E/2) + (Verror/2*4), where VP is phenotypic variance, VG is genotypic variance, VG × E is the variance of G × E interaction, and Verror e is variance error.

## 5. Conclusions

A genome-wide association approach was employed for the first time in *Ae. Biuncialis*, and revealed large genetic variation in grain edible fiber and protein content. BG, TP, and WEP content showed a high level of inheritance, making them promising traits to study a relationship between genotype and phenotype. Our GWAS analysis suggested that loci on chromosomes 1M^b^, 4M^b^, and 5M^b^ affect grain BG content, whereas chromosome groups 3, 6, and 1 contain a majority of QTLs responsible for TP and WEP content. *Aegilops* orthologs for genes found in these QTLs may be transferred from selected accessions and pyramided in bread wheat to achieve improved grain quality. The present study will facilitate more efficient use of *Ae. biuncialis* in pre-breeding programs to develop wheat varieties with specially modified fiber content and improved health benefits.

## Figures and Tables

**Figure 1 ijms-23-03821-f001:**
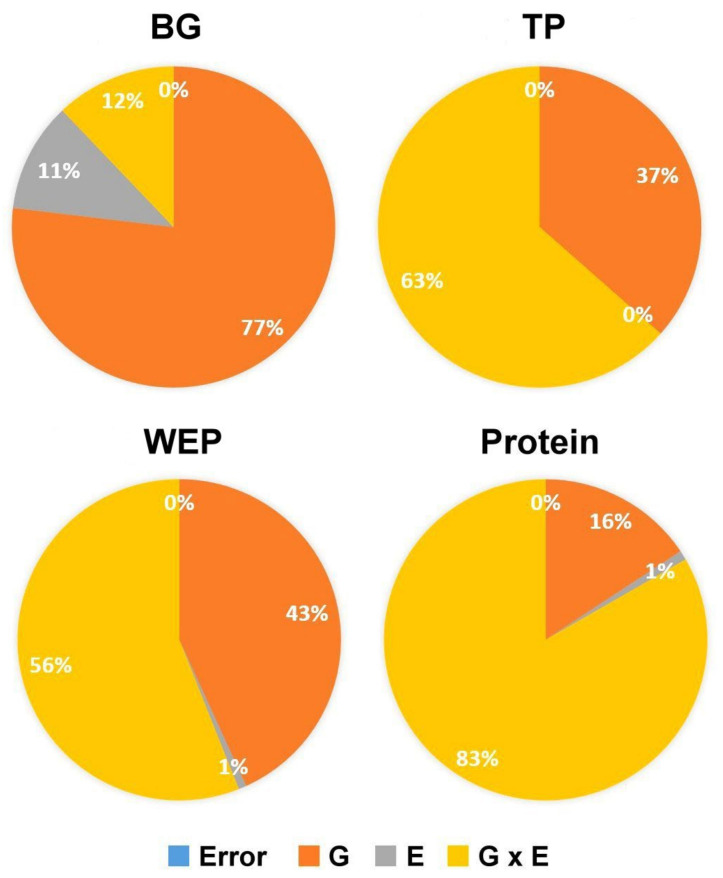
Variance components for grain compositional traits β-glucan (BG), arabinoxylan, measured as total- (TP) and water-extractable pentosans (WEP), and protein content in the *Ae. biuncialis* collection (*n* = 83).

**Figure 2 ijms-23-03821-f002:**
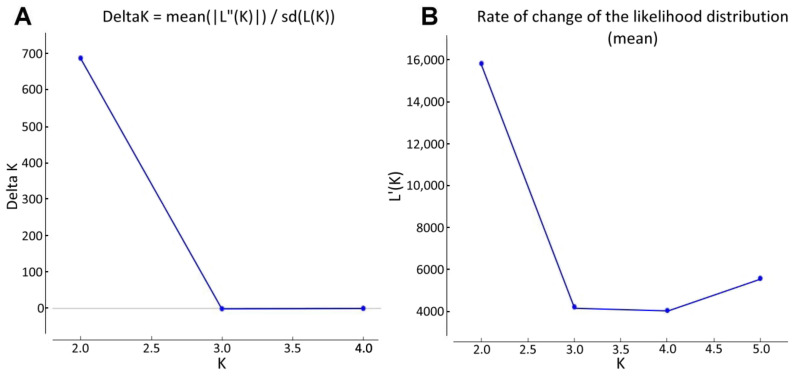
Plot of Delta K values (**A**) and the likelihood distribution (**B**) from the Structure analyses of *Ae. biuncialis* accessions, obtained through STRUCTURE Harvester web version (*n* = 86).

**Figure 3 ijms-23-03821-f003:**
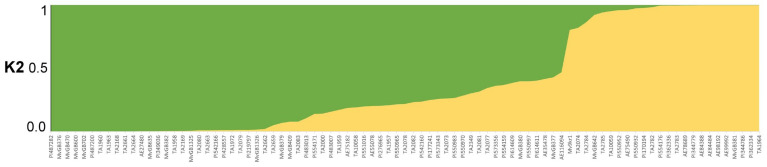
STRUCTURE bar plot for K = 2 based on genotyping data of DArTseq-derived SNP markers. Q value represents the proportion of ancestry to a given subpopulation.

**Figure 4 ijms-23-03821-f004:**
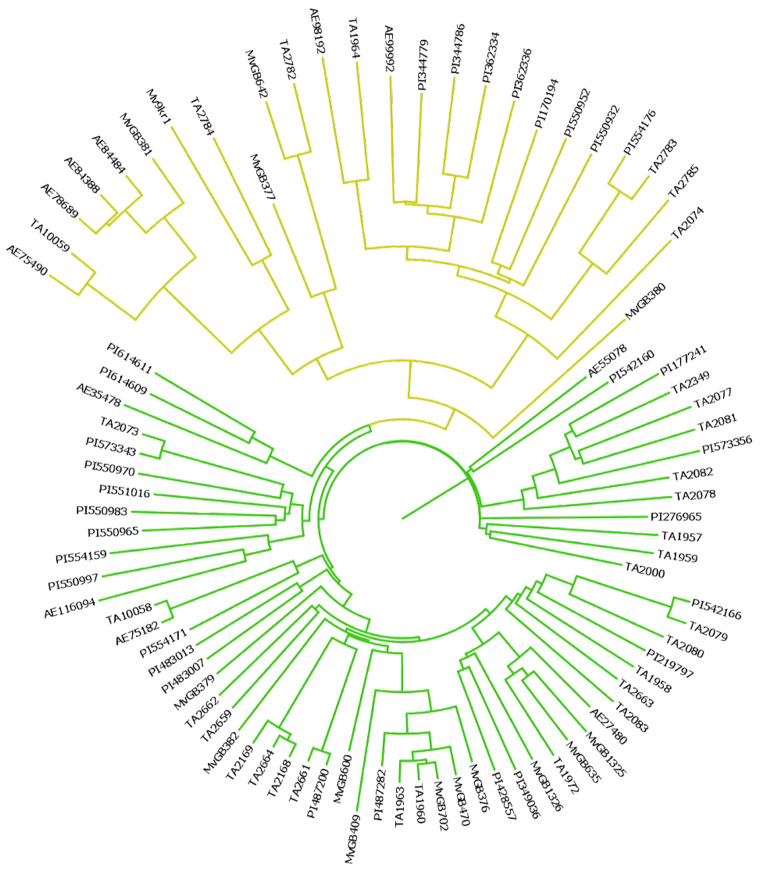
Unrooted Bayesian tree of the *Ae. biuncialis* collection created with the bootstrap method and 1000 replications on the data set of 2602 SNPs.

**Figure 5 ijms-23-03821-f005:**
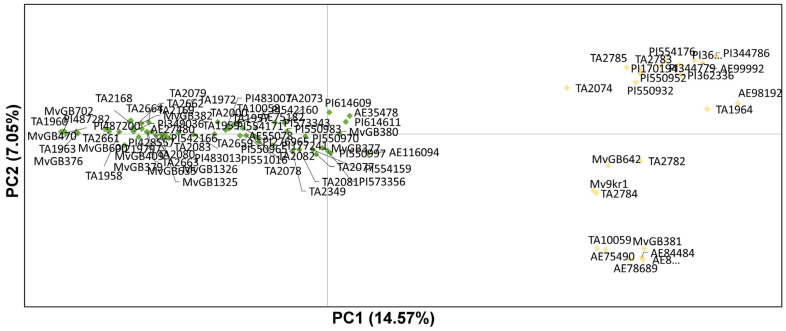
Principal Coordinate Analysis (PCoA) plot of the first two components obtained from the DArTSeq array of 2602 SNPs for 86 *Aegilops* accessions. The first two coordinates explained the 14.57% and the 7.05% of variability, respectively, for a total of 21.62%.

**Figure 6 ijms-23-03821-f006:**
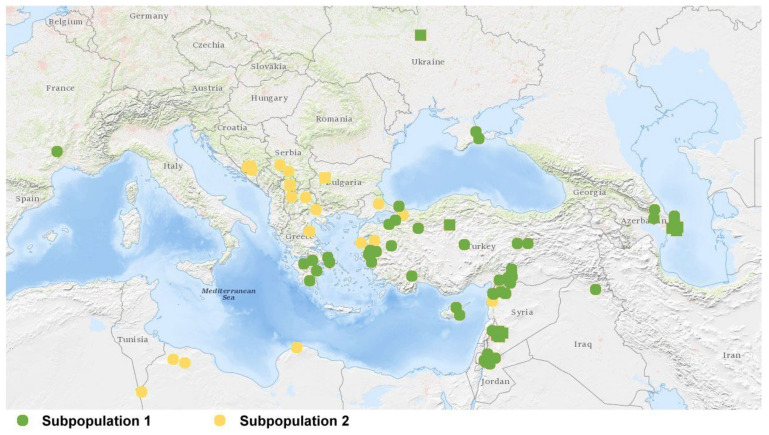
Geographic distribution of *Ae. biuncialis* accessions. Subpopulation 1 is represented by green symbols, while subpopulation 2 is represented by yellow symbols, as determined by three statistical methods. Circles represent accessions with known geographic locations. If only the country of origin is known, a square representing the capital is used (no information was available for six accessions).

**Table 1 ijms-23-03821-t001:** The content of β-glucan (BG), total- (TP) and water-extractable pentosans (WEP), and protein content in grains of *Ae. biuncialis* collection and Mv9kr1 wheat line grown in Martonvásár in 2016 and 2017.

	2016				2017			
	Mv9kr1	*Ae. biuncialis* Collection			Mv9kr1	*Ae. biuncialis* Collection		
		Mean *	Min	Max		Mean *	Min	Max
BG (mg/g)	9.44	38.10	22.70	54.90	8.62	35.30	19.50	51.20
TP (mg/g)	40.17	40.11	29.60	50.77	40.24	40.40	31.90	50.71
WEP (mg/g)	10.79	10.82	6.83	15.42	10.63	10.68	7.25	15.46
Protein (%)	12.91	26.61	19.61	33.49	13.01	27.20	22.80	33.00

*: *n* = 83 *Ae. biuncialis* accessions.

**Table 2 ijms-23-03821-t002:** The effect of genotype (G), environment (E), and their interaction (G × E) on the compositional quality traits of the *Ae. biuncialis* genotypes as determined by linear mixed model analysis.

	Mean Squares			h^2^
	G	E	G × E	
BG	244.6 ***	1366.5 ***	17.7 ***	0.93
TP	131.7 ***	13.4 ^n.s.^	61.7 ***	0.54
WEP	24.8 ***	3.04 **	3.7 ***	0.61
Protein	27.7 ***	59.3 ***	3.8 ***	0.27

**, ***: significant at 0.01 and 0.001 probability level, *n* = 83, ^n.s.^: not significant.

**Table 3 ijms-23-03821-t003:** Marker-trait associations, DArTSeq markers, marker locations on the chromosomes of *Aegilops* (UM) and wheat (ABD), and p values on chromosomal regions associated with dietary fiber trait components [β-glucan (BG), total-pentosan (TP) and water-extractable pentosan (WEP)], and grain protein content. GWAS analysis was performed for each individual year, considering all four quality traits estimated for both years as a covariate, only significant QTLs in both environments were reported.

Trait	QTL	Marker	Effect	Chr (UM) ^a^	Chr (ABD) ^b^	R^2^	*p*	LOD	Candidate Annotation
BG	1	100022501_F_0	−0.88	4M/6U	4ABD	0.128	6.87 × 10^−4^	4.542	glutathione S-transferase 3-like ^†^
	2	100013840_F_1	−0.78	5M	5ABD	0.125	7.72 × 10^−4^	3.113	-
	3	100079925_F_0	−1.62	1M **/1U	1ABD	0.149	2.26 × 10^−4^	3.647	-
Protein	4	100001630_F_1	−3.64	6M	-	0.124	8.39 × 10^−4^	3.076	putative ripening-related protein 6 ^†^
	1	100022501_F_2	−5.76	4M/6U	4ABD	0.159	1.30 × 10^−4^	3.886	glutathione S-transferase 3-like ^†^
	5	100011893_F_1	4.67	2M/2U	2ABD	0.139	3.68 × 10^−4^	3.435	-
	6	100074730_F_0	−1.67	1M **/1U	1ABD	0.140	3.54 × 10^−4^	3.451	-
	7	100016211_F_2	−8.13	2M/6U	2ABD	0.189	2.53 × 10^−5^	4.597	-
	8	100030135_F_1	−2.43	5M	5ABD	0.125	7.84 × 10^−4^	3.106	DNA-binding transcription factor activity *
	9	100054424_F_0	1.33	2M/2U	2ABD	0.126	7.44 × 10^−4^	3.128	-
	10	100033763_F_0	−2.13	4M/6U	2A 6B 4D	0.122	9.34 × 10^−4^	3.030	-
	11	100016524_F_0	−3.64	7M/7U	7ABD	0.145	2.78 × 10^−4^	3.556	-
	12	100024379_F_1	3.85	3M/3U	3ABD	0.138	3.92 × 10^−4^	3.407	-
	2	100013840_F_0	−3.10	5M	5ABD	0.147	2.41 × 10^−4^	3.618	-
	13	100013808_F_0	5.39	5M/5U	5ABD	0.123	8.56 × 10^−4^	3.068	DNA-binding transcription factor activity *
TP	5	100011893_F_2	0.10	2M/2U	2ABD	0.125	8.01 × 10^−4^	3.096	-
	7	100016211_F_0	−12.70	2M/6U	2ABD	0.184	3.44 × 10^−5^	4.464	-
	8	100030135_F_0	−6.40	5M	5ABD	0.144	2.96 × 10^−4^	3.529	DNA-binding transcription factor activity *
WEP	4	100001630_F_0	−1.67	6M	-	0.145	2.80 × 10^−4^	3.554	putative ripening-related protein 6 ^†^
	14	100009067_F_0	−3.16	1M/1U	1ABD	0.131	5.85 × 10^−4^	3.233	1-deoxy-D-xylulose-5-phosphate synthase activity *
	1	100022501_F_1	−2.54	4M/6U	4ABD	0.221	4.29 × 10^−6^	5.367	glutathione S-transferase 3-like ^†^
	15	100027188_F_0	−3.55	7M/7U	7ABD	0.140	3.58 × 10^−4^	3.446	-
	5	100011893_F_0	−0.33	2M/2U	2ABD	0.125	7.55 × 10^−4^	3.122	-
	16	100015451_F_0	4.12	1M **	1ABD	0.183	3.50 × 10^−5^	4.456	-
	7	100016211_F_1	−3.60	2M/6U	2ABD	0.198	1.54 × 10^−5^	4.812	-
	17	100013669_F_0	3.51	3M/3U	3ABD	0.149	2.19 × 10^−4^	3.660	-
	18	100033114_F_0	−2.29	7M7U	7ABD	0.126	7.37 × 10^−4^	3.133	DNA-binding transcription factor activity *
	19	100006546_F_0	−0.76	6U	6ABD	0.155	1.63 × 10^−4^	3.787	-
	20	100030958_F_0	4.47	6M	6ABD	0.123	8.59 × 10^−4^	3.066	glycosyltransferase At5g20260 *
	21	100041833_F_0	−2.45	-	-	0.132	5.55 × 10^−4^	3.256	-
	22	100001948_F_0	2.88	3M/3U	3ABD	0.131	5.84 × 10^−4^	3.234	soluble starch synthase *
	23	100010676_F_0	3.93	1M **/4U	6A 1B 1D	0.126	7.34 × 10^−4^	3.135	glycoside hydrolase/deacetylase superfamily *
	24	100001383_F_0	−2.25	3M/3U	3ABD	0.135	4.68 × 10^−4^	3.330	O-acetyltransferase activity *
	25	100070301_F_0	−1.83	2M/2U	2ABD	0.120	9.91 × 10^−4^	3.004	protein serine/threonine phosphatase activity *
	12	100024379_F_0	3.15	3M/3U	3ABD	0.161	1.17 × 10^−4^	3.934	-
	26	100036161_F_0	2.05	3M/3U	3ABD	0.123	8.75 × 10^−4^	3.058	-
	27	100009019_F_0	3.06	6M/4U	6ABD	0.144	2.95 × 10^−4^	3.530	glutamate receptor 2.8-like ^†^
	28	100024849_F_0	1.93	6M/6U	6ABD	0.127	6.97 × 10^−4^	3.157	endoglucanase 5-like ^†^
	29	100010471_F_0	4.48	6M	6ABD	0.123	8.79 × 10^−4^	3.056	-
	30	100005191_F_0	−3.51	4M/1U	5A 4B 4D	0.148	2.36 × 10-^4^	3.626	flavonol synthase/flavanone 3-hydroxylase-like ^†^
	31	100006760_F_0	−2.46	3M	3ABD	0.129	6.28 × 10^−4^	3.202	-
	32	100014148_F_0	−2.45	3M/3U	3ABD	0.161	1.21 × 10^−4^	3.919	-
	33	100066878_F_0	3.09	4M/6U	4ABD	0.142	3.21 × 10^−4^	3.493	-
	34	100069132_F_0	−2.00	1M **/6U	1ABD	0.166	9.09 × 10^−5^	4.041	zinc ion binding *

^a^ Location of QTLs on chromosomes belonging to the U and M genomes of diploid *Ae. umbellulata* and *Ae. comosa*, respectively. ^b^ Location of QTLs on chromosomes belonging to the A, B, and D genomes of hexaploid wheat. ***** Putative genes and enzymes influencing the grain quality traits were identified using UniProt and Pfam databases. ****** First and second-best hits on chromosomes 4M and 1M were similar, but previous syntenic data suggested 1M as a potential chromosomal location. ^†^ Genes described in the CAZy (Carbohydrate-Active enZYmes) database.

## Data Availability

Not applicable.

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
