# Peer review of "Identification of New QTLs for Dietary Fiber Content in Aegilops biuncialis"

_ijms, 2022, doi:10.3390/ijms23073821_

Round 1
Reviewer 1 Report
The research paper by Ivanizs et al. performed a genome-wide association study to dissect the genomic regions and candidate genes associated with dietary fiber content in Aegilops biuncialisl. However, I have some crucial criticisms/comments (detailed below):
Concerns:
- As per the manuscript, the authors did phenotyping for only two years/environment. But to identify the stable and reliable QTLs, one should perform analysis using multi-location/year phenotypic data (at least three years or locations). So, in my opinion, this data is less to conclude any results.
- Details of experimental design (randomized complete blocks design (RCBD), alpha lattice, etc.) used in this study were not provided.
- It is not clear, what is the –log10 P-value considered for marker-trait associations, please provide more details.
- The authors do not perform linkage disequilibrium (LD) analysis. LD is important as it allows identifying genetic markers that tag the actual causal variants. Further, more details on candidate gene identification should be provided.
- The discussion section should improve. They have to discuss their results and compare them with some earlier QTLs associated with dietary fiber content.
Author Response
"Please see the attachment."

Reviewer 2 Report
The manuscript submitted by Ivanizs et al. is an interesting contribution to the genetic control of dietary fiber content in Aegilops biuncialis and was well written. Using association analysis and DArTseq-derived SNP genotyping, they identified 34 QTLs associated with β-glucan, pentosan, water-extractable pentosan, and protein content in a set of 83 Aegilops biuncialis Vis. Accessions associated from a wide range of ecological habitats. Since I consider this number of population to be very small for GWAS analyses, especially for cereals, I wonder if the authors could explain the reason for selecting only a small number of 83 (86) accessions in their study. To improve the manuscript, I would like also to suggest the following corrections and changes in the order of presentation in the text:
1- Line 149-150 could go to M&M and further explain the model used for ANOVA analysis.
2- Line 162, please add the Mean square values to Table 2.
3- Line 188, please remove the K=3 and K=4 graphs from Figure 3.
4- Line 221, please add Manhattan plots and QQ plots obtained from your GWAS analysis with TASSEL as supplementary figure.
5- Line 227, did you use the FDR test or the LOD-3 test to detect a significant association between markers and traits? this is not enouph clear in the manuscript.
6- Line 233, please add the effect of each QTL and the physical location of the markers in based on the wheat genome in Table 3.
7- Line 235, it is better to write "GWAS analysis" instead of "QTL analysis".
8- Line 334, your work to identify candidate genes based on SNP positions is very speculative, especially in the case where there is not even a genetic map for your population and you could not find a QTL interval in your analysis. I recommend shortening this section and focusing only on the most meaningful results to be discussed here.
9- Line 367 and 372, "We identified" instead of "We mapped".
10- Line 462, you did not use a replicated experimental design in your study. Do you mean sites/locations instead of biological replications?
Author Response
"Please see the attachment."

Round 2
Reviewer 1 Report
The authors have clarified most of the questions I raised in my previous review. Unfortunately, a major problem (multilocation trial) has not been addressed by this revision. Now I feel the manuscript could be accepted for publication.
Author Response
"Please see the attachment."

Reviewer 2 Report
The manuscript was revised well. I only see one missing information in the text:
In the new version, you have added Fig. S3 for your LD analysis. The Figure needs to be elaborated in the Results and LD decay should be discussed in the manuscript appropriately.
Author Response
"Please see the attachment."
